

# Pre-industrial Temperature Variability on the Swiss Plateau Derived from the Instrumental Daily Series of Bern and Zurich

Yuri Brugnara[1,2], Chantal Hari[1,3,4], Lucas Pfister[1,2], Veronika Valler[1,2], and Stefan Brönnimann[1,2]

[1]Oeschger Centre for Climate Change Research, University of Bern, Switzerland
[2]Institute of Geography, University of Bern, Switzerland
[3]Wyss Academy for Nature, University of Bern, Switzerland
[4]Climate and Environmental Physics, Physics Institute, University of Bern, Switzerland

**Correspondence:** Yuri Brugnara (yuri.brugnara@giub.unibe.ch)

**Abstract.** We describe the compilation of two early instrumental daily temperature series from Bern and Zurich, Switzerland, starting from 1760 and 1756, respectively. The series are a combination of numerous small segments from different observers at different locations, within and outside the two cities, converted to modern units and homogenized. In addition, we introduce a methodology to estimate the errors affecting daily and monthly mean values derived from early instrumental observations. Given the frequent small data gaps, we merge the two daily series into a more complete series representing the central Swiss Plateau. We finally compare the homogenized monthly series with other temperature reconstructions for Switzerland. We find significant differences before 1860 pointing to biases affecting some of the most widely used instrumental data sets. In general, the homogenization of temperature measurements at the transition between the early instrumental and the national weather services eras remains a problematic issue in historical climatology and has significant implications for other fields of climate research.

## 1 Introduction

Meteorological early instrumental observations are usually defined as those measurements made before the creation of national weather services (NWSs) (Brönnimann et al., 2019a). Particularly in Europe, a wealth of early instrumental data have allowed scientists to reconstruct the climate variability of the pre-industrial era (e.g. Böhm et al., 2010; Dobrovolnỳ et al., 2010; Valler et al., 2021), although many more have never been used.

Early instrumental data are generally considered of lesser quality, mainly because of the lack of standard procedures in place before the centralization of station networks, particularly for temperature measurements. Nevertheless, global climate data sets offer hundreds of temperature records derived from early instrumental measurements. In many cases, these are data that were elaborated by scientists in the 19th or early 20th century (e.g. Dove, 1839; Blodget, 1857; Eredia, 1912) and have never been re-evaluated since. Even though the data were produced by excellent climatologists, the tools and the needs of 19th century scientists were quite different from today and there is still much scientific potential to be explored.

Another shortcoming of many early instrumental data series is their scarce traceability: it is often very hard, if not impossible, to trace the source of a publicly available record and the processing that it underwent, let alone to know how the underlying





data were measured. This lack of transparency is becoming more and more a relevant issue, in a time when climate data are as
25  crucial as never before.

In Switzerland the NWS (today: MeteoSwiss) was created in 1863, but the first regular instrumental measurements date
back to the early 18th century. The history of the oldest temperature measurements in Switzerland is shaped by the success
of Jacques-Barthélemy Micheli du Crest's "universal" thermometer, a spirit thermometer invented by the Genevan scientist in
1741 that became the most common thermometer in many of the Swiss cantons until the 1770s. In spite of the fact that the
thermometric liquid used was not mercury, Micheli du Crest's scale was more unambiguously defined than its main alternative
at the time, Réaumur's, facilitating the conversion to modern units (see Brugnara et al., 2020).

The best known and documented Swiss early instrumental temperature records are those of Basel (Bider et al., 1958), starting
in 1755, and Geneva (Schüepp, 1961), starting in 1768 or 1753 (when extended using data from Neuchâtel). Due to the very
fragmented records and frequent relocations, building consistent early instrumental series for Bern and Zurich brings higher
hurdles. Some versions of those series were included in several global and regional data sets (e.g. Auer et al., 2007), however
many parts are missing, while others are hard to identify.

During the last decade a lot of progress was made in digitizing the many existing Swiss early instrumental records (Füllemann
et al., 2011; Pfister et al., 2019; Brugnara et al., 2020). This large effort allows us to produce for the first time long and nearly
complete temperature series for the two main cities on the central Swiss Plateau, Bern and Zurich, reaching back to the mid-
18th century. Our goal is not only to provide useful data to the scientific community, but also to guarantee full data traceability
down to the original sources.

We focus on daily average temperature and also provide a tentative estimation of its uncertainty, despite the lack of metadata
for many of the records. As reference series for the homogenization we use raw (i.e, non-homogenized) data from nearby
stations, which allow us to produce results that are largely independent from existing temperature reconstructions.

The paper is structured as follows: in Sect. 2 we describe in detail the many data sources; in Sect. 3 we explain the methods
for the calculation of daily means, the homogenization, the merging of the different segments, and the data filling; in Sect. 4
we analyze the homogenized data and compare them with existing temperature reconstructions for Switzerland, before giving
our conclusions in Sect. 5.

## 2  Data

We started from raw data digitized from numerous weather diaries and publications (Pfister et al., 2019; Brönnimann, 2020).
The original temperature readings were expressed in the Micheli du Crest, Réaumur, and Celsius scales. The data conversion
to modern units followed the equations given in Brugnara et al. (2020). In addition to the data quality checks described in that
work, we performed systematic visual checks and corrected several minor digitization and conversion errors.

We use temperature measurements from at least 29 different locations (14 in Bern, 15 in Zurich) made between 1756–1867.
Moreover, we use data from nearby towns (Büren an der Aare, Burgdorf, Sutz, Küsnacht, and Winterthur) to fill gaps in the



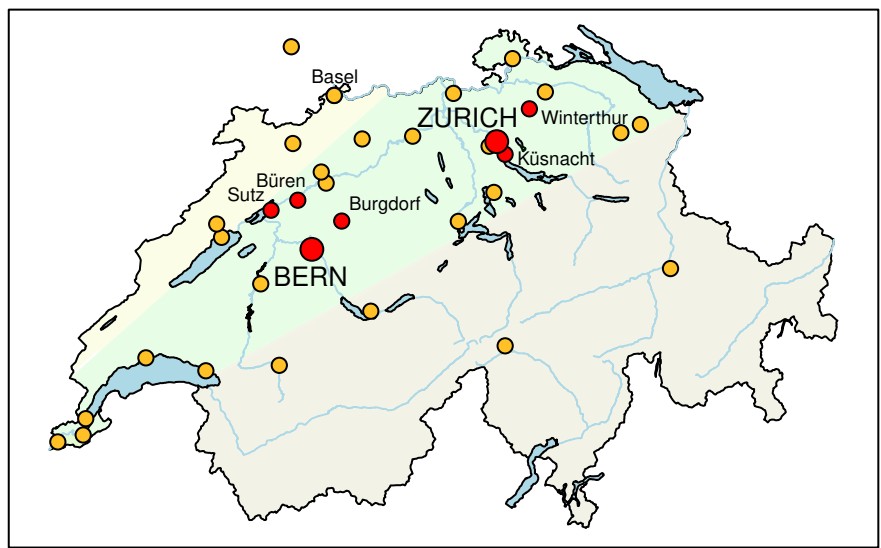

**Figure 1.** Map of Switzerland with the stations used in the homogenization (red: stations contributing to the Bern or Zurich series, gold: reference stations). The background colors outline the three main geographical regions of Switzerland (from top to bottom: Jura Mountains, Swiss Plateau, and Swiss Alps). Political borders and water bodies were provided by the Federal Office of Topography swisstopo.

two merged temperature series. In total over 300,000 data points contribute to the two merged series (169,600 for Bern and 137,884 for Zurich). All records also include pressure measurements, which are not considered in this study.

In this section we briefly describe each contributing data source and provide the relevant references for additional informa-tion. In general, temperature was measured outside a window facing north, typically 5 to 10 m from the ground and with no 60 sheltering from radiation. Significant differences from this "standard" setting are mentioned in the source's description when known. All times refer to local time.

We use additional raw data from Brugnara et al. (2020), from the DigiHom projects (Füllemann et al., 2011), and newly rescued data from Basel and Geneva (Brönnimann et al., 2022; Brugnara and Brönnimann, 2022; Brugnara et al., 2022) as reference series for the homogenization (Fig. 1; a detailed list is provided in the Supplement). To evaluate the homogenized 65 series, we compare them with the HISTALP version 3 (Auer et al., 2007; Böhm et al., 2010) and EKF400 version 2 (Valler et al., 2021) gridded data sets. Both products are based on the same homogenized monthly temperature series (including data from Bern and Zurich), although they follow rather different approaches: HISTALP is a simple interpolation of homogenized instrumental temperature series, while EKF400 is a paleo-reanalysis that assimilates multiple variables (temperature, pressure, precipitation) as well as documentary and proxy records (e.g., tree rings) into a global model simulation, providing a 30- 70 member ensemble of possible realizations.



**Table 1.** Records contributing to the daily temperature series of Bern (elevations are estimated)

| No. | Location | Elevation (m) | Observer | Period | Rank (1–4) | Contribution (%) |
|---|---|---|---|---|---|---|
| 1 | Bern, Monbijou | 532 | Tavel | 1760–1766 | 1 | 6.7 |
| 2 | Bern, Burgerspital | 540 | Lombach | 1777–1785 | 1 | 8.5 |
| 3 | Bern, Burgerspital | 540 | Studer | 1779–1789 | 2 | 0.4 |
| 4 | Bern, Salzmagazin | 540 | Lombach | 1785–1789 | 1 | 4.5 |
| 5 | Sutz | 447 | Sprüngli | 1785–1802 | 3 | 1.0 |
| 6 | Büren | 440 | Studer | 1789–1797 | 2 | 6.8 |
| 7 | Bern, Old City | 540 | Studer | 1797–1801 | 1 | 4.7 |
| 8 | Bern, Old City | 540 | Studer | 1801–1803 | 1 | 1.8 |
| 9 | Bern, Old City | 540 | Studer | 1803–1827 | 2 | 22.4 |
| 10 | Bern, Villette? | 540? | Fueter | 1803–1806 | 3 | 0.0 |
| 11 | Bern, Villette? | 540? | Fueter | 1819–1833 | 3 | 0.3 |
| 12 | Bern, Old City | 548 | Trechsel | 1826–1846 | 1 | 20.8 |
| 13 | Bern, Old City | 542 | Benoit | 1837–1853 | 2 | 3.3 |
| 14 | Bern, Old City | 542 | Trechsel | 1848–1849 | 1 | 1.8 |
| 15 | Bern, Bollwerk | 551 | Wolf | 1851–1855 | 1 | 4.1 |
| 16 | Burgdorf | 538 | Fankhauser | 1851–1863 | 3 | 2.6 |
| 17 | Bern, Old City | 546 | Koch | 1855–1858 | 1 | 3.1 |
| 18 | Bern, Old City | 585 | Reinhard | 1860–1863 | 1 | 3.1 |
| 19 | Bern, Observatory | 574 | Simler, NWS | 1863–1867 | 2 | 4.1 |

## 2.1 Bern

To build the Bern series we use records that were digitized during the CHIMES project (Brugnara et al., 2020), with minor integrations from previously overlooked sources. The records are summarized in Table 1 and Fig. 2. In addition, Figure 3 provides an overview of the time coverage of each record.

### 2.1.1 Tavel, Lombach, and the Bern Economics Society (1760–1789)

The first meterological station was established in 1760 by the Bern Economics Society (Ökonomische Gesellschaft Bern, hereafter ÖGB) as part of a network of stations in the region of Bern, one of the very first in history to use standardized instruments and measurement practices (Pfister, 1975). Franz Jakob "Monbijou" von Tavel (1729–1798) was the observer designated by the ÖGB for the station of Bern . Himself a prominent member of the ÖGB and of its meteorological commission, he started the observations in January 1760 in his estate, a few hundreds meters southwest of the city gate in what is todays



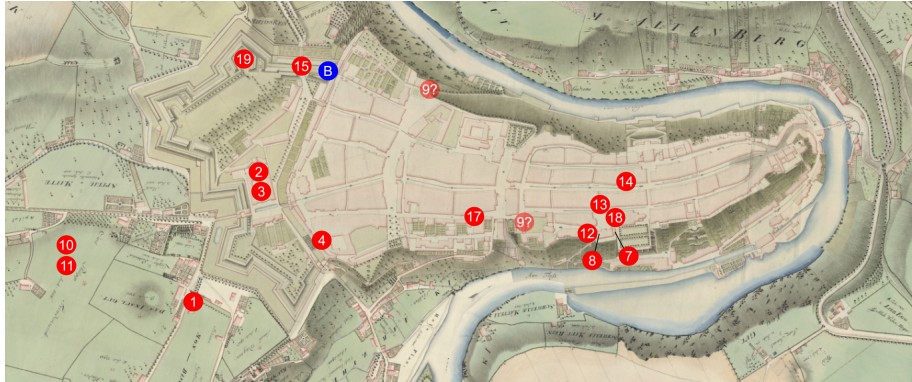

**Figure 2.** Measurement locations drawn on a historical map of Bern (J.R. Müller, ca. 1797). The numbers refer to Table 1, the "B" indicates the position of the MeteoSwiss station of Bern-Bollwerk. Map source: Geodaten Stadt Bern, Amt für Geoinformation des Kantons Bern.

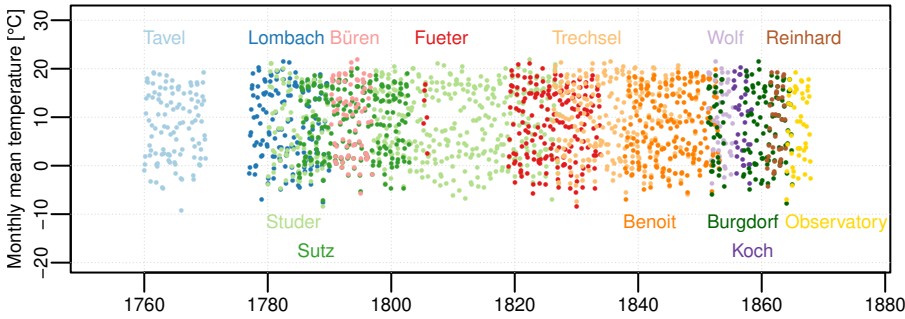

**Figure 3.** Monthly means of the raw data used to build the Bern series with observers' names or locations.

Monbijoustrasse. At that time the area was not urbanized. More detailed information on this record can be found in Wyer et al. (2021).

The network of the ÖGB initially used mercury thermometers with Réaumur scale. A comparison with other stations (Wyer et al., 2021) revealed that this was a so-called "false" Réaumur thermometer (see Camuffo, 2020), meaning that the scale was calibrated between the freezing and boiling point of water and can be transformed to Celsius through a constant factor. However, in March 1762 the instrument in Bern was replaced with a Micheli du Crest spirit thermometer, which requires more complex conversion formulas (see Brugnara et al., 2020).

The observation times recommended by the ÖGB were at sunrise and 3:00 PM (Pfister, 1975). We assumed that these were used by the station in Bern, although no explicit time is given in the data source. Tavel made an additional observation in the evening, which we assumed to be at 10:00 PM after a comparison with the modern diurnal cycle in Bern and with other stations. However, it is very likely that the observation times were not followed strictly, not least because of the low accuracy of clocks in the 18th century (see e.g. Camuffo et al., 2021).





The measurements were temporarily interrupted between July 1766 and March 1767 due to family issues (Pfister, 1975). Unfortunately, the data for the last three years (1767–1769) are only available as monthly means. The ÖGB eventually lost
interest in the network and the observations were discontinued in 1770.

A few year later, the ÖGB reactivated a meteorological station in the city. In January 1777 Karl Lombach (1740–1811) started the observations at the Burgerspital hospital (where he worked as a secretary – the building still exists today). Lombach initially used a Micheli du Crest thermometer but replaced it with a Deluc-type thermometer (equivalent to a "false" Réaumur) in January 1778. All observers who started measuring after him used this type of instrument.

The observation times were written down explicitly every day and varied depending on the season. Most of the time Lombach carried out two observations – one in the early morning and one in the afternoon – in line with the recommendations of the ÖGB. He added a third observation in the evening only in case of unusual events, such as during extreme cold spells.

Observations temporarily stopped between March and June 1785. The reason is probably a station relocation. In fact, a note on the sheet for June 1785 in a different handwriting reads "at the salt depot", corresponding approximately to todays address
of Bundesgasse 20, only 150 m to the south-east of the previous location. Observations in May and June 1787 are also missing. An analysis of Lombach's data is provided in Hari et al. (2022).

### 2.1.2 Studer (1779–1827)

Samuel Studer (1757–1834) was the priest of the Burgerspital and himself a member of the ÖGB as well as co-founder of the Swiss Natural Sciences Society (Schweizer Naturforschende Gesellschaft, hereafter SNG). In December 1779, on his own
initiative, he started making regular temperature measurements at the same location of Lombach (but in a different apartment). He measured three times per day at variable times that he always wrote down.

In December 1789 Studer moved to Büren an der Aare, a small town located about 20 km north of Bern (and at 100 m lower elevation), where he was assigned the role of pastor for the local parish. He continued his measurements there until January 1797, when he returned to Bern to take the position of professor of theology at the Hohe Schule (the predecessor of todays
University). During this time, he lived in at least three different apartments. His meteorological measurements cover nearly half a century, until July 1827. An analysis of Studer's data is provided in Hari et al. (2022).

In at least two of Studer's apartments there was no suitable position with northern exposure for the thermometer. To make up for that, he installed two or sometime three thermometers with different exposures, taking care of marking the observations that were affected by direct sunlight (we excluded those observations). We mostly use the measurements from his primary
thermometer (i.e., the first value that he wrote down) except for the period after 1803, when we combine the afternoon and evening observations from the primary thermometer (eastern exposure) with the morning observations from the secondary thermometer (western exposure).

There is much uncertainty on where Studer lived and measured after 1803. On the one hand, he wrote that he lived "at the school", which may be interpreted as the main building of the Hohe Schule on the southern side of the Old City. On the other
hand, he also mentioned being close to the botanical garden (Hari, 2021), which at that time was located north of the Old City.



Studer was a dedicated naturalist and travelled often to the Alps. As a consequence, his meteorological observations have frequent short gaps, particularly in summer.

### 2.1.3 Fueter (1803–1833)

Samuel Emmanuel Fueter (1775–1851) was a tradesman handling in colonial goods. He is remembered mainly for being the
first importer of tea in Bern. Very little is known about his meteorological measurements. Most likely he measured at his family's estate just outside the city (the Villette, roughly corresponding to todays Kocher Park), not far from where Tavel used to measure. This is supported by the particularly low temperatures reported in the morning, suggesting a rural environment.

We recovered two segments of his temperature observations: 1) December 1803 to November 1806 (with many gaps) and 2) January 1819 to November 1833. The first segment contains irregular observations, mostly once per day. In the second
segment the observations are taken much more regularly – twice per day at sunrise and 2:00 PM. An analysis of Fueter's data is provided in Hari et al. (2022).

### 2.1.4 Trechsel, Benoit, Wolf, Koch, Reinhard, and the Natural Sciences Society of Bern (1826–1863)

In April 1826 the Natural Sciences Society of Bern (Naturforschende Gesellschaft in Bern, hereafter NGB) established its own weather station in the city. The designated observer was the former president of the Society, Friedrich Trechsel (1776–1849),
who measured at his house in front of Bern's Cathedral. The observation times were fixed at 9:00 AM, 12:00 PM, 3:00 PM, and 10:00 PM. The evening observation was anticipated to 9:00 PM from February 1844 onward. In January 1848 Trechsel moved to a nearby house at the address Kramgasse 12, less than 100 m from the Cathedral, where he continued the measurements until September 1849. Unfortunately, the data for the year 1847 could not be found. In addition, Trechsel was often away from Bern in late summer, with nobody taking over the measurements in his absence.

Daniel Gottlieb Benoit (1780–1853) was also a member and former president of the NGB, but carried out his meteorological measurements as an independent amateur after leaving the NGB in 1832. He was a neighbour of Trechsel, living and measuring on the opposite side of the Cathedral's square. Benoit measured twice a day at 6:00 AM and 2:00 PM, between 1837–1853. More detailed information on Trechsel's and Benoit's records can be found in Flückiger et al. (2020).

Trechsel was followed by Johann Rudolf Wolf (1816–1893), an astronomer, best known today for his work on sunspots. He
was also a meteorologist and became in 1863 the first director of the Swiss NWS. Between 1851–1855, while director of the astronomical observatory in Bern, he measured temperature at his apartment, at the feet of the hill where the observatory stood (the Grosse Schanze). Wolf measured seven times per day and was the first observer in Bern to use the Celsius scale.

When Wolf moved to Zurich in May 1855, his assistant Johann Rudolf Koch (1832–1891) took over both the job and the measurements. According to Gimmi et al. (2007), Koch measured in todays Amthausgasse in the old town. The thermometer
was hung about 10 m from the ground with northeast exposure. Koch measured every 4 hours from 8:00 AM to 8:00 PM. His observations were the first published by the NGB, although only weekly means were published for the first 6 months. Luckily, the daily means were compiled by Wolf when he was director of the NWS and could therefore be recovered. Koch carried out





temperature observations until May 1858; the publication of the other meteorological variables (pressure, precipitation, etc.), however, continued until June 1860, suggesting that they were made at a different location (perhaps the observatory).

The successor of Koch at the direction of the observatory was Heinrich von Wild (1833–1902), one of the most influential meteorologists of the late 19th century. In 1860 he hired the guardian of the Cathedral's bell tower, Johann Reinhard, to continue the measurements on behalf of the NGB. The station was now part of a public network financed by the Canton of Bern (Hupfer, 2017), similar in scope to the network created one century earlier by the ÖGB. In the meantime, the astronomical observatory was refitted to host self-registering instruments that would serve for the upcoming NWS.

Reinhard measured from November 1860 until November 1863 at 7:00 AM, 2:00 PM, and 9:00 PM. Wild reported that the thermometer needed a correction of -0.2 K (Wild, 1862), which we applied. The thermometer was hosted inside an elaborated metallic radiation screen, a significant innovation for the time. Wild estimated that the use of the screen reduced the radiative bias to only 0.3 K on average, when installed at least 3 m from the ground on a north-facing wall (Wild, 1860). This screen would become a standard in some parts of Europe during the following decades (e.g. K.K.-Central-Anstalt, 1893; Nordli et al.,
1997). It eventually evolved into the so-called "Wild screen" (Auchmann and Brönnimann, 2012), a free-standing metallic radiation shield in use until the 1970s.

Reinhard's apartment was located inside the bell tower, about 50 m above the ground. Any radiation bias was probably further reduced by the abundant ventilation at that height. On the other hand, we expect a smaller diurnal temperature range at such an exposed location, which is confirmed by a comparison with the other records.

It is not clear when systematic temperature measurements at the astronomical observatory on the Grosse Schanze began. According to Wild (1862), a self-registering thermograph was installed latest during the winter 1861/62 and placed inside a wooden shelter on the northern corner of the observatory's terrace. However, we could only find tabulated data by Wild's assistant Rudolf Theodor Simler (1833–1873) starting from June 1863. The station was officially incorporated in the Swiss national network on 1 December 1863 and continued operating until 1898, when it was relocated outside of the city. We use
the data until December 1867 as reference against which all other records are homogenized.

### 2.1.5   Other Stations

Despite the many early instrumental records available for Bern, there remains gaps not covered by any segment, mainly between 1770–1776 (1766–1776 at daily resolution), 1789–1796, and 1858–1860. In addition to the already mentioned Büren an der Aare (Sect. 2.1.2), we use data from Sutz (1785–1802) and Burgdorf (1851–1863) to fill part of these gaps.

Sutz lies 25 km northwest of Bern, on the southern shore of Lake Biel at about 100 m lower elevation than Bern. The observer was Johann Jakob Sprüngli (1717–1803), a pastor. He observed three times a day – the exact times are not given but are probably rather consistent (Pfister, 1975) – using a Micheli du Crest thermometer.

Burgdorf is located less than 20 km to the northeast in a setting similar to that of Bern (hilly landscape at similar elevation). The observer in Burgdorf was Rudolf Ludwig Fankhauser (1796–1886), also a pastor. The style of his observations is very
similar to that of Sprüngli: he also measured three times a day without specifying the exact time, except for the afternoon observation at 2:00 PM.



For both stations we set the observation times to be at sunrise, 2:00 PM, and 9:00 PM, which are typical times for early instrumental observers. The uncertainty introduced by this assumption will be discussed in the methods.

We considered filling the gap between 1766–1776 with data from Neuchâtel and Gurzelen (Brugnara et al., 2020), both
stations being relatively close to Bern, but we eventually renounced to do so. The record of Neuchâtel is problematic because of the frequent relocations and travels of the observer Frédéric Moula, while that of Gurzelen was measured indoor. Moreover, the climate of these stations is slightly different from that of Bern, being the former more influenced by Lake Neuchâtel and the Jura mountains, the latter by the Alps.

## 2.2 Zurich

The records that we used to build the Zurich series are described in detail in Fritze et al. (2021) and Brugnara et al. (2021b) and are summarized in Table 2 and Fig. 4. In addition, Figure 5 provides an overview of the time coverage of each record.

### 2.2.1 18th century: Ott, Meyer, Muralt, Hirzel, and the Physical Society

The earliest instrumental observations in Zurich date back to 1708 (Brugnara et al., 2021a). However, temperature measurements for the first half of the 18th century are either unusable or lost. In the late 1750s Zurich saw an explosion of weather
measurements among learned individuals, at a time where the thermometer conceived by Micheli du Crest was gaining popularity in many of the Swiss cities.

The earliest record that we use is by Johann Jakob Ott (1715–1769), a correspondent of Micheli du Crest and Lambert, who measured several times per day (at irregular times) at his estate north of the city (Rötel), from July 1756 until his death. Being head of the Meteorological Commission of the Physical Society of Zurich, he certainly had a strong influence on the three
observers that started measuring in the city of Zurich shortly after him: Hans Conrad Meyer (1693–1766), Daniel von Muralt (1728–1793), and Hans Caspar Hirzel (1725–1803).

Meyer measured at the old hospital, of which he was the master, between 1759–1765. We could recover the original measurements in graphical form between June 1761–December 1762, taken four times daily at fixed times. For the remaining years until 1764 we found daily averages calculated by Wolf from the morning and evening observations.

Muralt was a merchant. He measured three times daily at his house in todays Bahnhofstrasse, from 1760 until his death in 1793. We found the original measurements in graphical form for 1760–1769, 1781–1785, and 1787–1793. For the years 1770–1779 we could only find a transcription of the morning observations made between January–April. Exact times for afternoon and evening observations are not always given, but we assume that they are kept constant at 1:00 and 9:00 PM throughout the record.

Hirzel, a prominent member of the Physical Society, probably started measuring earlier than Ott, but we could only find sporadic measurements from 1759 and regular ones (usually thrice daily at variable times) for 1761–1762, 1767–1786, and 1795–1802. Until 1786 he measured in todays Glockengasse, later he moved to his wife's estate outside of the city (the exact location is not known). We are not certain whether the later years are really from Hirzel, as the data structure and the handwriting differ from earlier data.





**Table 2.** Records contributing to the daily temperature series of Zurich (elevations are estimated)

| No. | Location | Elevation (m) | Observer | Period | Rank (1–4) | Contribution (%) |
| --- | --- | --- | --- | --- | --- | --- |
| 1 | Zurich, Rötel | 460 | Ott | 1756–1769 | 4 | 3.3 |
| 2 | Zurich, Old City | 415 | Meyer | 1759–1764 | 1 | 5.2 |
| 3 | Zurich, Old City | 415 | Muralt | 1760–1793 | 2 | 19.1 |
| 4 | Zurich, Old City | 420 | Hirzel | 1761–1786 | 3 | 9.3 |
| 5 | Zurich, unknown | unknown | Hirzel | 1795–1802 | 1 | 7.6 |
| 6 | Zurich, Hirschengraben | 435 | Feer | 1807–1827 | 3 | 16.5 |
| 7 | Zurich, Old City | 430 | Horner | 1812–1821 | 4 | 0.1 |
| 8 | Zurich, Old City | 430 | Escher | 1816 | 4 | 0.0 |
| 9 | Zurich, Old City | 420 | Horner | 1823–1834 | 2 | 6.0 |
| 10 | Zurich, Old City | 415 | Nüscheler (SNG) | 1831–1832 | 3 | 0.3 |
| 11 | Zurich, Hirschengraben | 430 | Hofmeister? | 1834–1835 | 1 | 0.8 |
| 12 | Zurich, Hirschengraben | 430 | Ulrich, Hofmeister? | 1835–1842 | 1 | 7.4 |
| 13 | Zurich, Kantonsschule | 442 | unknown | 1842–1852 | 1 | 10.5 |
| 14 | Küsnacht | 420 | Zollinger and others | 1852–1856 | 2 | 3.8 |
| 15 | Winterthur | 443 | Furrer | 1857–1867 | 2 | 6.1 |
| 16 | Zurich, Old Botanical Garden | 420 | Brügger | 1860–1866 | 3 | 0.0 |
| 17 | Zurich, Observatory | 470 | NWS | 1864–1867 | 1 | 4.1 |

### 2.2.2 Early 19th century: Feer, Horner, Escher, and the Swiss Natural Sciences Society

The record by Johannes Feer (1763–1823), an engineer, is one of the longest available for Zurich, covering the period 1807–1827. Feer probably lived at the Hirschengraben, in the eastern part of the city, and measured two or three times per day at variable times. There are many short interruptions in the record, hinting at frequent travelling. Moreover, the year 1826 is entirely missing. Feer cannot actually have been the observer during the last few years, as he died in 1823.

Johann Kaspar Horner (1774–1834), professor of mathematics, started his measurements in 1812 in todays Florhofgasse. In 1823 he moved to todays Bahnhofstrasse, where he continued measuring until his death. Therefore, his record covers an even longer period than Feer's. However, the measurements are very discontinuous before 1823, so that only a handful of monthly means can be calculated in that period. Probably Horner was particularly interested in studying the diurnal cycle of temperature, as he performed sometime up to 16 measurements in a day. On average he took four observations per day at variable times.

Between 1826–1830 Horner sent his data to the SNG, which was trying to set up a national network of meteorological observatories. Starting from 1830 the SNG began to hire young students as temporary observers for Zurich, each measuring only for a few months at a new location. We only use the longest of these segments (February 1831–March 1832), which covers part of a 6-month gap in Horner's series. The observation times required by the SNG were 9:00 AM, 12:00 PM, and 3:00 PM.





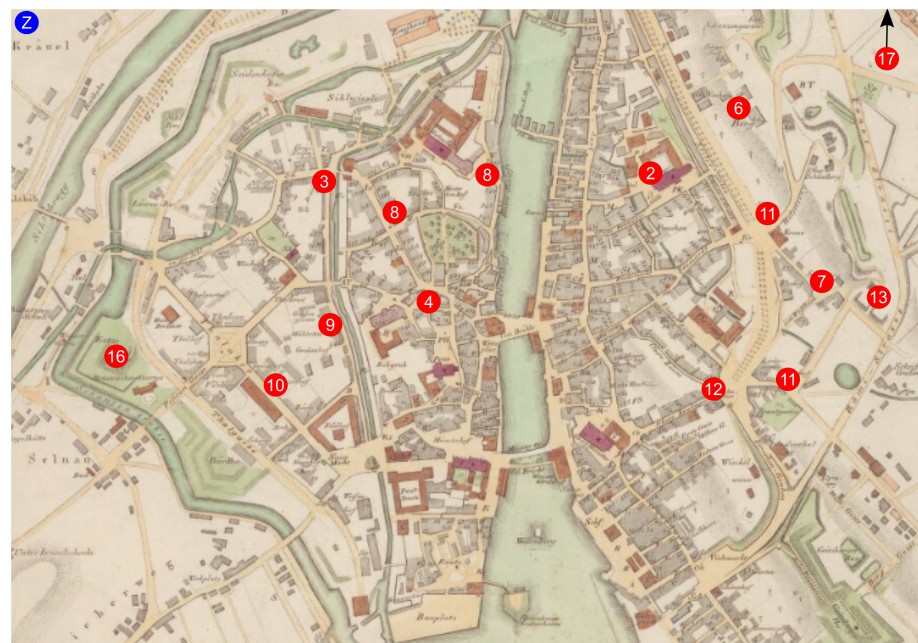

**Figure 4.** Measurement locations drawn on a historical map of Zurich (Heinrich Keller, 1838). The numbers refer to Table 2, the "Z" indicates the position of the MeteoSwiss station of Zurich-Zeughaushof. Note that the position of the Observatory (17) lies outside of the map. Map source: Staatsarchiv des Kantons Zürich.

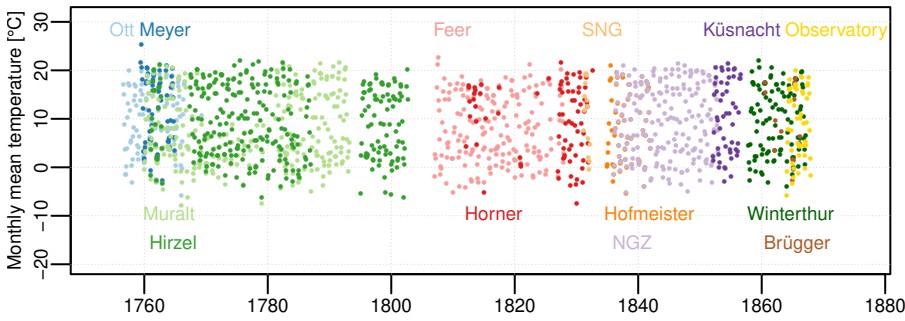

**Figure 5.** Monthly means of the raw data used to build the Zurich series with observers' names or locations.

We use an additional short record in this period by Johann Kaspar Escher (1744–1829), covering 1816–1820. However, we

could only find the monthly means published in Escher (1822), with the exception of the last 7 months of 1816, for which daily observations at 12:00 PM were published.





### 2.2.3   Hofmeister, Ulrich, and the Zurich Natural Sciences Society (1834–1852)

The Zurich Natural Sciences Society (Naturforschende Gesellschaft in Zürich, hereafter NGZ), successor of the Physical Society, published regular meteorological observations starting in 1836. The observer until 1842 was officially Melchior Ulrich
(1802–1893), professor of theology at the University. However, the instruments were spread over multiple locations and Ulrich was certainly not the only observer. We found handwritten data attributed to Rudolf Heinrich Hofmeister (1814–1887) that are mostly identical to those published by the NGZ but contain more frequent measurements (up to a dozen per day, of which 4 were published). Hofmeister's journal goes back to September 1834 and mentions a relocation to Ulrich's house (Hirschengraben) in August 1835.

In November 1842 the station was moved to the newly built cantonal school (Kantonsschule) on the eastern outskirts of the city, where it continued operating until 1852. The publication of the data, however, stopped in 1848, and for 1849–1851 we could only find daily averages (based on the morning and evening observations) published by the NWS (MCSNG, 1867).

### 2.2.4   The 1860s: Brügger and the Astronomical Observatory

After 7 years of gap, the next record that we could recover is by Christian Gregor Brügger (1833–1899), a natural scientist
known in meteorology for organizing during the 1850s a large network of weather observers in the canton of Grisons. After moving to Zurich in 1859, he started to measure at the old botanical garden, where he was the curator of the botanical collection of the Polytechnikum (todays Swiss Federal Institute of Technology or ETH). His measurements, however, are of rather poor quality (see Brugnara et al., 2021b) and very discontinuous.

With the creation of a NWS at the end of 1863, Zurich got its first official station at the NWS headquarters, in the newly built
astronomical observatory on the hill northeast of the old city. We include the first 4 years of the observatory's record (1864–1867) in our early instrumental series, in order to use it as the reference against which all previous records are homogenized.

### 2.2.5   Other Stations

To fill the long gap between 1853–1863 we use data from Küsnacht (1852–1856) and Winterthur (1857–1867). Küsnacht lies on the eastern shore of Lake Zurich; Winterthur is located about 20 km northeast of Zurich at similar elevation. We did not
include the station of Uetliberg (a hill west of Zurich) because of the large difference in elevation (over 400 m) and because of strong radiative biases affecting that record. Several gaps in the series of Zurich remain, in particular between 1793–1794, 1802–1806, 1832–1834, and 1856–1857 (Fig. 5). Records that would cover at least the most recent gaps probably exist but have yet to be found (see Pfister et al., 2019).

The record of Küsnacht was initiated by Heinrich Zollinger (1818–1859), a botanist known for being the first European
to climb Mount Tambora after the 1815 eruption. In 1848 he took the position of director at the seminary of Küsnacht after spending 6 years on the island of Java, where he would return in 1855. His measurements at the seminary were carried on by several individuals, most likely teachers or students. The measurements were taken every 4 hours between 6:00 AM and 10:00 PM.





We could not find any information about the observer of Winterthur, except that the family name was probably "Furrer". The
observation times were fixed at 9:00 AM, 12:00 PM, and 4:00 PM.

## 3 Methods

### 3.1 Daily Means

Daily means should be ideally calculated from continuous (e.g., hourly) measurements. It is common practice, however, to
calculate them as the arithmetic average of the daily maximum and minimum temperature. Neither approach is possible for
early instrumental observations, because self-registering instruments and max/min thermometers were not common until the
mid-19th century. On the other hand, a simple arithmetic average of the available observations would lead to obvious inhomo-
geneities and would not be always representative of the true daily mean.

A common approach is to apply a correction to the arithmetic average derived from a modern mean climatological diurnal
cycle of temperature (e.g., Moberg et al., 2002). The correction is different for each day, because it depends on the observation
times and the season. This approach works sufficiently well if the goal is to obtain homogeneous monthly means, but it will
likely produce an unaccurate statistical distribution of the daily means by overestimating day-to-day variability (Brandsma and
Können, 2006). To avoid this problem, we use instead a least-squares multiple linear regression (MLR) model defined as:

$$T_m = a_0 + \sum_{i=1}^{n+2} a_i x_i + \epsilon \tag{1}$$

where the predictors $x_i$ are the elements of the vector

$$\boldsymbol{x} = \left[ sin\left(\frac{2\pi j}{366}\right), cos\left(\frac{2\pi j}{366}\right), T_1, ..., T_n \right], \tag{2}$$

$T_m$ is the mean daily temperature (i.e., the predictand), $j$ is the Julian day, $n$ is the number of measurements (up to 6;
additional measurements are excluded) and $T_i$ the observed temperature values on the analyzed day, $a_i$ are the regression
parameters, and $\epsilon$ is the residual error. The first two elements of $\boldsymbol{x}$ are added to capture the seasonal variability of the diurnal
cycle. Both $T_m$ and $T_i$ are expressed in terms of anomalies with respect to a daily climatology calculated by fitting to the daily
means the first two harmonics of a Fourier series:

$$f(j) = b_0 + \sum_{k=1}^{2} \left[ b_k \cos\left(\frac{2\pi k j}{366}\right) + c_k \sin\left(\frac{2\pi k j}{366}\right) \right]. \tag{3}$$

The model is trained on sub-hourly data (10-minute resolution) from the MeteoSwiss urban stations of Bern-Bollwerk
(located on a roof at almost the same location where Wolf's apartment stood in the 1850s) and Zurich-Zeughaushof (located in
a courtyard close to the main train station), both covering the period 1991–2020. Modern-day urban heat island has certainly





**Table 3.** Performance of the MLR model compared to a correction based on the climatological diurnal cycle (CLIM) for common combinations of observation times (expressed in mean solar local time and 24-hour notation)

| Obs. times | RMSE (K) | | $\Delta$IQR (%) | |
|---:|---|---|---|---|
| | MLR | CLIM | MLR | CLIM |
| 7 | 1.76 | 1.80 | -20.6 | -12.7 |
| 14 | 1.29 | 1.78 | -4.0 | 35.3 |
| 21 | 1.17 | 1.26 | -8.9 | 1.9 |
| 6, 14 | 0.55 | 0.63 | -1.8 | -1.4 |
| 8, 20 | 0.55 | 0.59 | -5.6 | -5.4 |
| 7, 14, 21 | 0.28 | 0.41 | -1.6 | 1.4 |
| 9, 12, 15 | 0.60 | 0.94 | -2.0 | 14.6 |
| 8, 12, 16, 20 | 0.33 | 0.69 | -1.6 | 7.5 |
| 9, 12, 15, 22 | 0.42 | 0.72 | -1.2 | 8.6 |
| 6, 10, 14, 18, 22 | 0.17 | 0.41 | -0.9 | 1.8 |

an effect on the diurnal cycle at these stations (e.g., Gubler et al., 2021). Nevertheless, they provide the best available estimate of the climatological diurnal cycle in the early instrumental period.

In Table 3 we show the results of a validation of our MLR approach based on the data from Bern-Bollwerk, in which we use the first 6 years (1991–1996) for validation and the rest for training the model. Our validation metrics are the root mean squared error (RMSE) and the relative bias of the interquartile range ($\Delta$IQR) in percent, defined as:

$$\Delta IQR = \frac{IQR_{predicted} - IQR_{true}}{IQR_{true}}. \tag{4}$$

For comparison we also show the performance of the standard approach based on the climatological diurnal cycle, calculated from 31-day windows centered on each calendar day.

The MLR clearly outperforms the standard approach with respect to the RMSE, particularly when three or more measurements are taken. It tends to slightly underestimate the IQR but it is more consistently close to the true value for different
combinations of observation times, whereas the standard approach can lead to a large overestimation particularly when there are measurements in the afternoon.

However, we still use the mean climatological adjustments instead of the MLR for those data that are only available as daily means. This is the case for 6 months in the Bern series (Koch) and 7 years in the Zurich series (Meyer and NGZ).

We set the start of each day at midnight Greenwich Meridian Time (GMT), consistently with the modern-day convention
followed by MeteoSwiss. Original times are converted to GMT using mean solar local time (GMT+00:30 for Bern, GMT+00:35 for Zurich) until 1847 and Bern Time (GMT+00:30) afterwards. Monthly means are calculated from daily means following WMO (2008).





## 3.2 Error Estimation

Given the large heterogeneity affecting early instrumental records, an objective estimation of the error is critical for the data
user. However, given the lack of metadata, it is not possible to quantify in detail each source of error affecting the measurements.
Some error estimates are necessarily approximative and/or based on educated guesses.

We consider five types of error related to: 1) reporting resolution ($e_1$); 2) number of measurements in a day ($e_2$); 3) time
uncertainty ($e_3$); 4) exposure ($e_4$); and 5) climate ($e_5$). The total standard error for daily means is then given by:

$$E_d = \sqrt{\sum_{i=1}^{5} e_i^2}. \tag{5}$$

The standard error for monthly means is:

$$E_m = \frac{1}{N} \sqrt{\sum_{i=1}^{N} E_{d(i)}^2}, \tag{6}$$

where $N$ is the number of non-missing daily means in the month.

### 3.2.1 Reporting Resolution

While modern thermometers employed in meteorology have a resolution of at least 0.1 K, the scales of early liquid-in-glass
thermometers had rarely a resolution higher than 0.5 degrees. Initially only spirit thermometers could attain a finer resolution,
thanks to the high thermal expansion of alcohol. Nevertheless, observers reported frequently with a resolution of 0.1 degrees
by extrapolating the scale visually.

We estimate the actual instrumental resolution $\delta$ from the frequency of fractional digits reported by the observer. To obtain
the resolution error in the daily mean we then scale with the number of measurements $n$ on a certain day:

$$e_1 = \frac{\delta}{2\sqrt{n}}. \tag{7}$$

### 3.2.2 Number of Measurements

In general, the more measurements are carried out in a day, the smaller the error of the daily mean will be. However, since the
measurements are made at different times by each observer, the number of measurements alone is not sufficient to estimate the
error. In addition, the number of measurements and the observation times can change from one day to another. Therefore, each
day can have a different error (see Tab. 3). As a measure of this error we simply take the standard deviation of the residuals $\epsilon$
of the MLR model (Eq. 1), aggregated by month.

Where the MLR could not be applied, we estimate the error from the differences between the true daily means in the modern
data and the means that would result from the measurement times at hand in the target day (corrected using the climatological
diurnal cycle).




### 3.2.3 Time

Before the development of railways in the mid 19th century, there was little need for a common time within a country. Each town had its own time, which roughly followed solar time (as measured by a sunclock). Mechanical clocks were not yet very accurate and long cloudy periods (particularly common on the Swiss Plateau) made sunclocks useless.

Taking meteorological observations at fixed times was thus far from trivial and probably this is one of the reasons why some observer did not report exact observation times. We can assume that observers relied on their daily routine, measuring at the times that were the most practical (e.g., shortly after waking up or before going to bed). However, they certainly also wanted their measurements to be as meaningful as possible. The best way to achieve that was to measure close to the coldest and warmest hours of the day, i.e. at sunrise and in the afternoon.

Our error estimation related to the precision of observation times is based largely on common sense and on the experience with observers such as Studer, who did not follow a fixed observation schedule but always noted the observation time, giving us an idea on how variable the observation times could be. We assign a constant standard error of 90 minutes to observation times that are inferred (e.g., Sutz), 45 minutes to those noted once per month at most (e.g., Benoit), 30 minutes to those noted each day (e.g., Studer), and 15 minutes to any non-inferred time after 1848 (the year of the introduction of the Bern time, GMT+00:30, in all of Switzerland). These errors include the conversion error caused by the variable difference between mean and apparent solar time, which ranges between 0–16 minutes (Camuffo et al., 2021).

The time error is transformed empirically into a daily mean temperature error from the mean diurnal cycle at Bern-Bollwerk and Zurich-Zeughaushof. For example, the error for the record of Tavel (inferred times) ranges between 0.1 K in December and 0.6 K in July.

### 3.2.4 Exposure

Unscreened thermometers are very sensitive to radiation, their exposure is thus fundamental for the reliability of the measurements. Guidelines on how to set up a thermometer already existed in the early instrumental era, but were not very detailed. Some general rules (e.g., northern exposure), however, remained fairly consistent over the period covered by our data (e.g., Carrard, 1763; Wild, 1860).

An unshielded thermometer hung on a north-faced wall is subjected to radiative biases from direct and indirect radiation that result in a temperature overestimation when compared to a modern weather station, particularly in the early morning and late afternoon (Böhm et al., 2010). This bias strongly depends on the exact orientation of the building, the surrounding environment, and the season.

Several factors, however, can interfere with these expectations: for example missing shadows from roofs or trees can lead to a stronger radiative bias in winter or early spring. In addition, not all observers had a north-facing window available (e.g., Studer), making biases more unpredictable. In some cases a wrong estimation of the observation time could result in an apparent radiative bias.



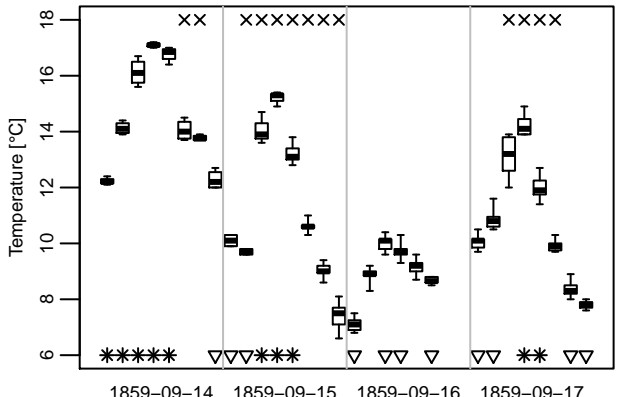

**Figure 6.** Results of the comparison described in Wild (1860). The boxes give the temperature range measured by up to seven thermometers in different positions, every 2 hours between 8:00 AM and 10:00 PM. The symbols indicate weather conditions observed by Wild (asterisk=sunshine, triangle=rain, cross=wind).

Unlike other systematic errors in the data, radiative biases are difficult to correct at daily resolution because they strongly depend on weather conditions. While the mean radiative bias can be dealt with by the homogenization, the exposure error takes into account mainly the variations related to the weather.

We can rely on some literature based on data from the Alpine region to obtain a plausible estimation of the error. Already Wild (1860) tried to estimate the uncertainty related to the position of the thermometer, to make the case for the benefits of his radiation screen. He did so by hanging seven thermometers in slightly different positions and on different materials on the north-western and northern walls of the astronomical observatory in Bern, and by reading them every 2 hours between 8:00 AM and 10:00 PM. Even though he carried out the comparison only for 4 days (between 14–17 September 1859 – on the first 385 day he read only four thermometers), his results are still interesting.

As shown in Fig. 6, Wild measured differences that were usually smaller than 1 K. The largest ones were at noon, when the mean standard deviation of the measurements is of 0.5 K; the smallest were at 6:00 and 8:00 PM, with 0.2 K (note that sunset was shortly after 6:00 PM). Certainly these differences would have been larger in early summer and with more stable weather. However, the temperature range reached 1 K even on the 16th of September, a rainy day with overcast sky.

One of Wild's thermometers, the one hung directly on the north-facing wall at 5 m height, illustrates particularly well the uncertainty of unscreened temperature measurements related to weather conditions. On the first day, a sunny day with little wind, it was the warmest thermometer at noon with 0.6 K above the average of all thermometers. On the second day, again sunny but windy, the same thermometer was the coldest, measuring 0.5 K below the average at noon. On the third day, rainy and calm, it was still 0.3 K below the average. Finally, on the fourth day, overcast and windy, it stood at 1.1 K below the 395 average. Therefore, an exposure compatible with early instrumental guidelines can lead to a standard error at noon of the order of 1 K, at least in summer.





In the modern literature, Böhm et al. (2010) analyzed a parallel record at Kremsmünster, Austria, finding that exposure is more relevant between May–August and can cause temperature biases from -2 to +5 K. Given the north-northeast exposure, the most affected time at that station is the early morning, when biases cover a range of about 4.5 K in July (1st–99th percentiles).
In winter, biases range between $\pm 1$ K, with no preferred time. Assuming normally distributed biases, this corresponds to a standard error for sub-daily observations of about 0.4 K.

In our data we can take advantage of several parallel records to estimate the impact of different exposures. For example, the standard deviation of daily mean differences between the Cathedral's tower and the astronomical observatory in Bern (both located in an elevated position and at similar elevation) was 1.1 K during November 1863 – the only month with comparable
observation times. For 2:00 PM observations we find the largest standard deviation in July 1863 with 1.0 K, compatible with a standard error of 0.7 K at both stations. For comparison, the standard error for a modern (non-urban) station is of the order of 0.1 K (Brandsma and Van der Meulen, 2008).

Based on these considerations, we defined a rather simple but plausible error function for a northern exposure that depends on the Julian day ($j$) and the number of measurements per day ($n$), but not explicitly on the observation times:

$$e_4 = \frac{1}{\sqrt{n}} \left( a + b \sin \frac{2\pi(j - 81)}{N} \right) \tag{8}$$

where $a = 0.8$ K (average error), $b = 0.4$ K (amplitude), and $N$ is the number of days in the year. The error has a minimum at the winter solstice and a maximum at the summer solstice. For $n = 3$, it ranges between 0.23–0.69 K. When $n > 3$, we use the same error obtained for $n = 3$ in order to avoid too small errors when $n$ is large.

We use different values of the parameters in Eq. 8 for the following records:

– Studer (incl. Büren): we estimate $a$ and $b$ from the available parallel measurements (given that the error function does not depend on the observation time, we assume that both of Studer's thermometers have the same exposure error), with the exception of 1801–1803, when the primary thermometer had a northern exposure;

– Reinhard, Bern Observatory: we estimate $a$ and $b$ from the available parallel measurements at 2:00 PM between the two records (June to November 1863), assuming that they have identical errors;

– Tavel, Wolf, Koch, Burgdorf, Meyer, NGZ, Küsnacht, Zurich Observatory: we inflate the error by 50% ($a = 1.2$ K, $b = 0.6$ K) in those months for which we found evidence of a strong radiative bias.

To identify strong radiative biases we analyzed the mean deviation of the sub-daily temperature measurements from the expected diurnal cycle of the respective months (see e.g. Brugnara et al., 2021b). The diurnal cycle was calculated from the modern data of Bern-Bollwerk and Zurich-Zeughaushof, to which 2 K were subtracted to account for climate change.
In addition, we took into account information from metadata. In fact, it was not uncommon for the observer to write about expected radiative biases in the measurements, although Wild was the only one to provide a quantitative estimation.

All different versions of the exposure error for Bern are summarized in Fig. 7. As expected, exposure errors in the Studer's measurements are generally larger, while they are smaller and less season-dependent at the stations that used radiation screens





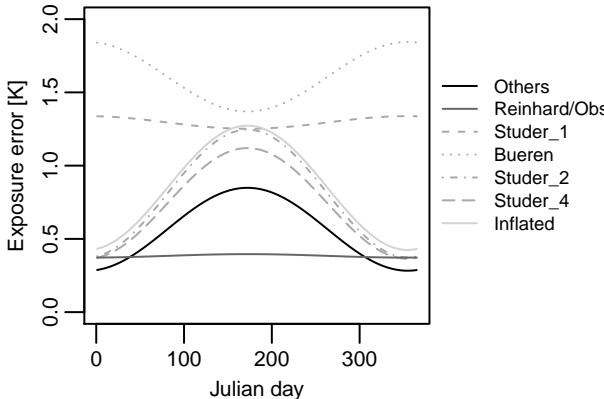

**Figure 7.** Comparison of different estimations of the exposure error $e_4$ for $n = 2$.

(Reinhard and Observatory). The maximum error is not always in summer, which is compatible with eastern or western expo-

sure.

### 3.2.5   Climate

For the records measured far from the city center of Bern (Büren, Sutz, and Burgdorf) and Zurich (Rötel, Küsnacht, and Winterthur) we need an additional error that takes into account possible climatological differences (e.g., more or less frequent fog). To do that we compare the modern daily series from the city center with data from other MeteoSwiss stations that are

representative of the climate of these "outsiders". The error is given by the root-mean-square deviation calculated from daily mean anomalies, aggregated by month. For Küsnacht we only take half of the resulting error, because the closest representative MeteoSwiss station (Stäfa) is much further away from Zurich than Küsnacht is.

### 3.3   Homogenization

We detect inhomogeneities within each record visually using the Craddock test (Craddock, 1979). A list of detected inhomo-

geneities is provided in the Supplement. Station relocations are considered inhomogeneities a priori. To avoid using inhomogeneous periods as reference, we use the same method to split reference series into homogeneous segments.

We calculate monthly adjustments from data overlaps between records and from reference series from nearby stations (a detailed list of reference series is provided in the Supplement). We then take the median of the monthly adjustments obtained from each reference series and transform them into daily adjustments by fitting the first two harmonics of a Fourier series (see

Eq. 3).

For segments shorter than 12 months we simply apply a constant correction identical for all days. With the exception of Mulhouse, all reference stations are from within the borders of Switzerland (see Fig. 1). The number and quality of reference series decreases rapidly before ca. 1780 and so does the accuracy of the adjustments. The data are adjusted backward in time



starting with the most recent segment, with respect to which all others all adjusted. The whole homogenization process is based
exclusively on raw data, therefore our results are not influenced by previous work.

### 3.4    Prioritization and Merging of the Daily Data (Swiss Plateau Series)

The many overlaps between different records imply that we need some criteria to choose, on a given day, which record to
use. We introduced a subjective quality ranking (from 1, best, to 4, worst) based on our knowledge of the data and metadata
(homogeneity, exposure, representativity, completeness, etc.).

Due to the frequent occasional missing data in most records, it is common that a monthly mean in the merged series is
calculated from daily means of different records (when more than one exists). On the other hand, a daily mean and its error are
always calculated from one and the same record.

To reduce the quantity of missing data in the daily series we produce a combined series given by the average of the Bern
and Zurich merged series. Missing days in one of the series are filled by adjusting the values in the corrisponding days of the
other series, using constant monthly adjustments calculated from the difference between the two homogenized series. We call
this combined series the Swiss Plateau temperature series.

### 3.5    Monthly Infilling and Merging with the MeteoSwiss Series

Remaining isolated data gaps in the monthly series are reconstructed from nearby stations using a weighted average, where
the squared Pearson correlation coefficients are the weights (Alexandersson and Moberg, 1997). Each reference series is first
brought to the same average of the series to be filled using the overlapping period (done separately for each month). Reference
series are checked for homogeneity and only homogeneous segments are used.

Only partially missing years are filled. Twelve monthly values are reconstructed for Bern (1% of the series), 109 for Zurich
(9%), and 14 in the Swiss Plateau series (1%).

Finally, in order to analyze climate variability over the last 265 years, we also merge our early instrumental series with
monthly temperature series for Bern and Zurich for the period 1864–2021 homogenized by MeteoSwiss (Begert et al., 2005).
For this we apply further adjustments to our data based on the mean monthly differences in the overlapping period (1864–1867).

## 4    Results and Discussion

### 4.1    Errors

Figures 8 and 9 show the evolution in time of the error of daily means for the series of Bern and Zurich, respectively. On
average the total error is 1.16 K for Bern and 1.07 K for Zurich. The mean error difference between June and December is
0.69 K and 0.82 K, respectively. The largest contributions to the total error come from the number of measurements ($e_2$) and
the exposure ($e_4$). The climate error ($e_5$) is also an important contributor in the segments affected by it. Total errors larger than
2 K affect 7.7% of the Bern series (mostly the Büren segment) and 2.1% of the Zurich series.

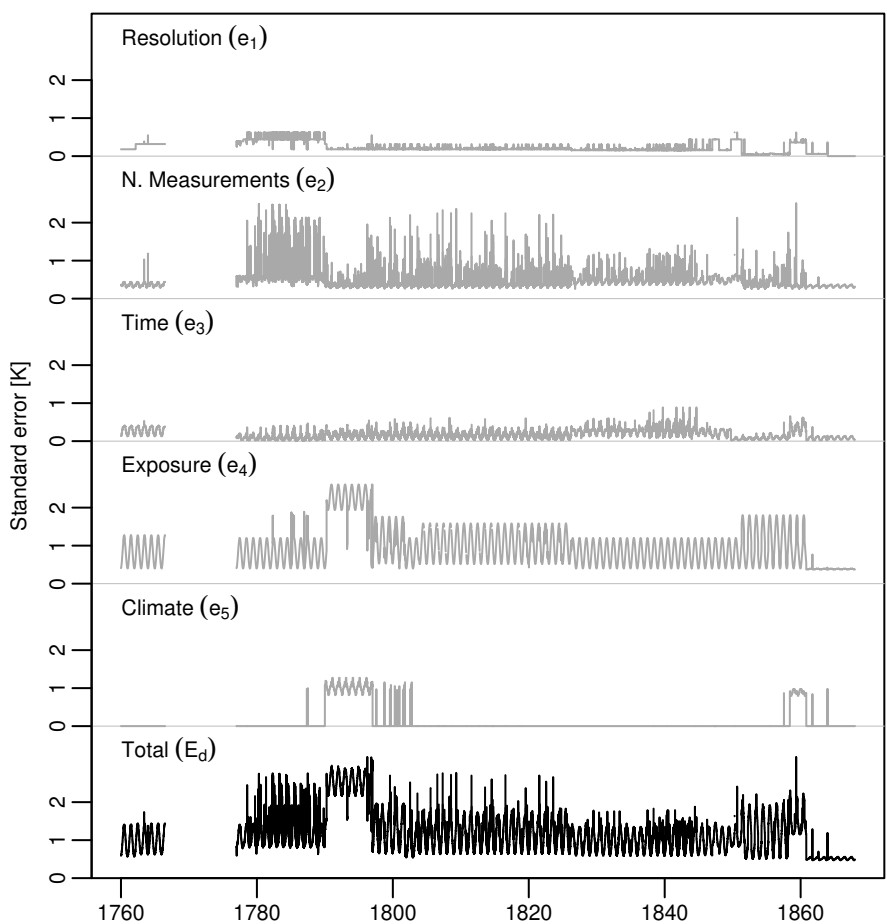

**Figure 8.** Time evolution of the daily standard error and its components for the merged Bern series.

The standard error of monthly means is on average 0.20 K and is never larger than 0.53 K (not shown). These values are
smaller than existing estimates for early instrumental data (e.g., Valler et al., 2021).

## 4.2   Homogenization Assessment

Early instrumental measurements are affected by radiative biases, poor ventilation, and heat exchange with buildings, all
causing temperature overestimation on average. Consistently, the adjustments that we applied to the raw data to homogenize
the series are mostly negative (Fig. 10).
Zurich is 0.9 K warmer than Bern on average after the homogenization, whereas it was 0.5 K warmer in the raw data. For
comparison, the mean difference between the modern stations of Zeughaushof and Bollwerk is 1.0 K.





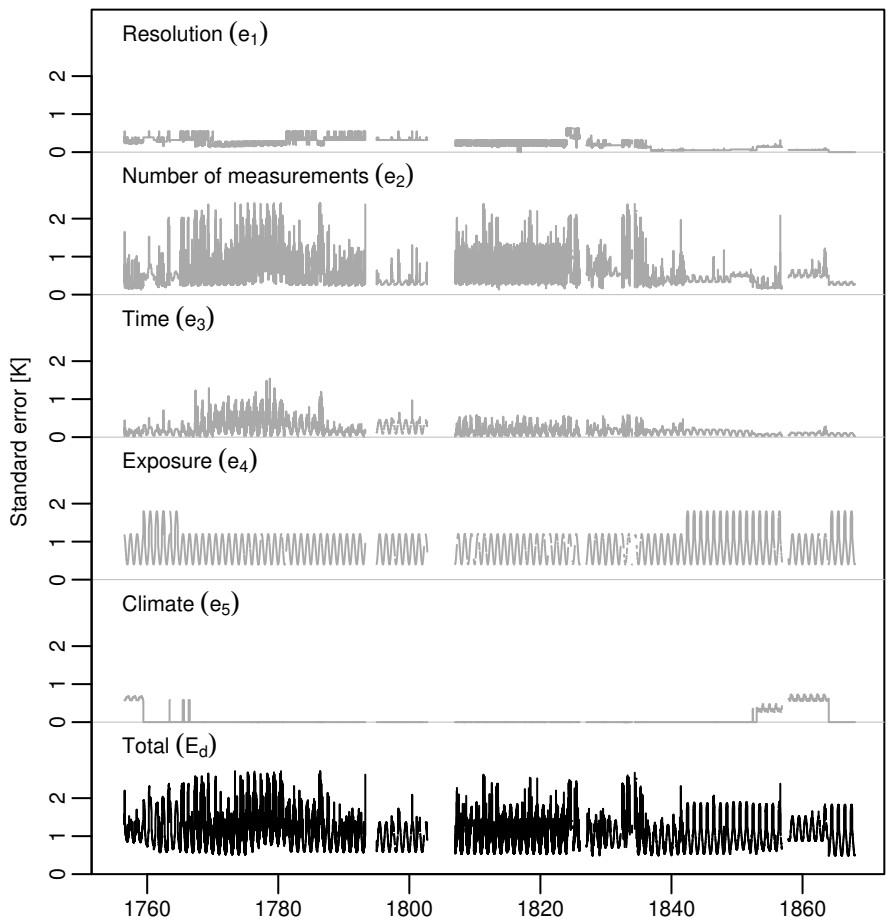

**Figure 9.** Time evolution of the daily standard error and its components for the merged Zurich series.

When comparing our homogenized early instrumental series with the closest grid point in the HISTALP and EKF400 data sets (Fig. 11 and 12), we obtain correlation coefficients between 0.86–0.96 for annual and seasonal averages. The lowest correlations occur in summer, when interannual variability is small and errors are large. Despite the generally high correlation,

a few noteworthy differences emerge, particularly in the last decade and at the turn of the century.

Both Bern and Zurich series are warmer than HISTALP after ca. 1855, more strikingly after 1860 when they show over 0.5 K higher mean annual temperature anomalies. The ensemble mean of EKF400 (which assimilates HISTALP) lies close to our data, hinting at a possible problem in HISTALP. However, it is possible that our series suffer from residual inhomogeneities in that period, too, especially considering the large fragmentation of data sources.

The warm bias shown by HISTALP between the 1790s and the early 1800s is another likely problem in that data set. Large differences with other data sets in that period were also mentioned in Böhm et al. (2010). Again, our series agree better with





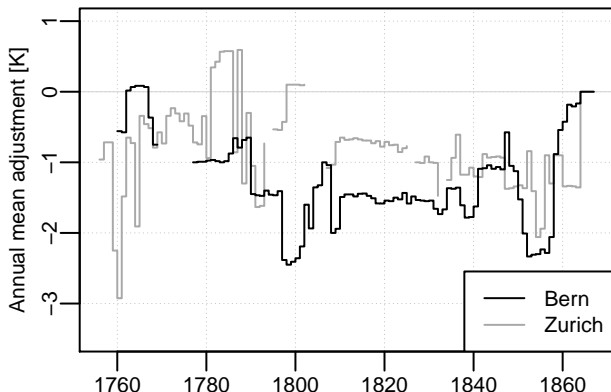

**Figure 10.** Time evolution of annual mean adjustments in the merged series.

the ensemble mean of EKF400. The differences come mainly from the warm season and might be related to residual radiative biases in HISTALP.

Extreme seasons such as the cold summer of 1816 or the cold winter of 1829/1830 are reproduced consistently by all data
sets, with some exception in early years, when the Zurich series is substantially warmer than the other series in winter. There is also good agreement on the warmest (1822 and 1834) and coldest (1785 and 1816) years.

A deteriorating quality of the homogenization can be expected before 1777 for Bern and before 1795 for Zurich, particularly in the seasonal means. The reason is the lower number of reference series with respect to later years, a problem further exacerbated by the large data gaps in our series. Note that both EKF400 and HISTALP are fully independent from our data
before 1777 for Bern and before 1830 for Zurich.

### 4.3 Daily Indices

The daily resolution of the data allows us to analyze daily temperature indices representing, for instance, the frequency of warm and cold days. Here we focus on the number of days with mean temperature above 20°C and below -5°C (Fig. 13). On the Swiss Plateau these thresholds represent moderately warm (cold) summer (winter) days. In the early instrumental period
they are exceeded on 17% of the days during June–August and December–February, respectively.

For both indices the 1810s stand out as an extremely cold decade. The "Year Without a Summer" 1816 (Brugnara et al., 2015) was not the only year with an anomalous summer in that decade; in fact, 1815 had even less warm days, although its summer mean temperature was considerably higher. The years 1812 and 1813 had also a very low number of warm days. At the same time Switzerland saw an unusually long sequence of harsh winters starting with 1808/1809 (see also Reichen et al.,
2022), followed by a sequence of relatively mild winters between 1817–1822.

A remarkable sequence of warm summers occurred in the early 1780s, mostly remembered by historians because of the coincidental Laki eruption in Iceland and the related optical and weather phenomena in Europe in 1783 (Zambri et al., 2019).

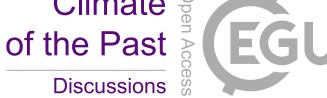



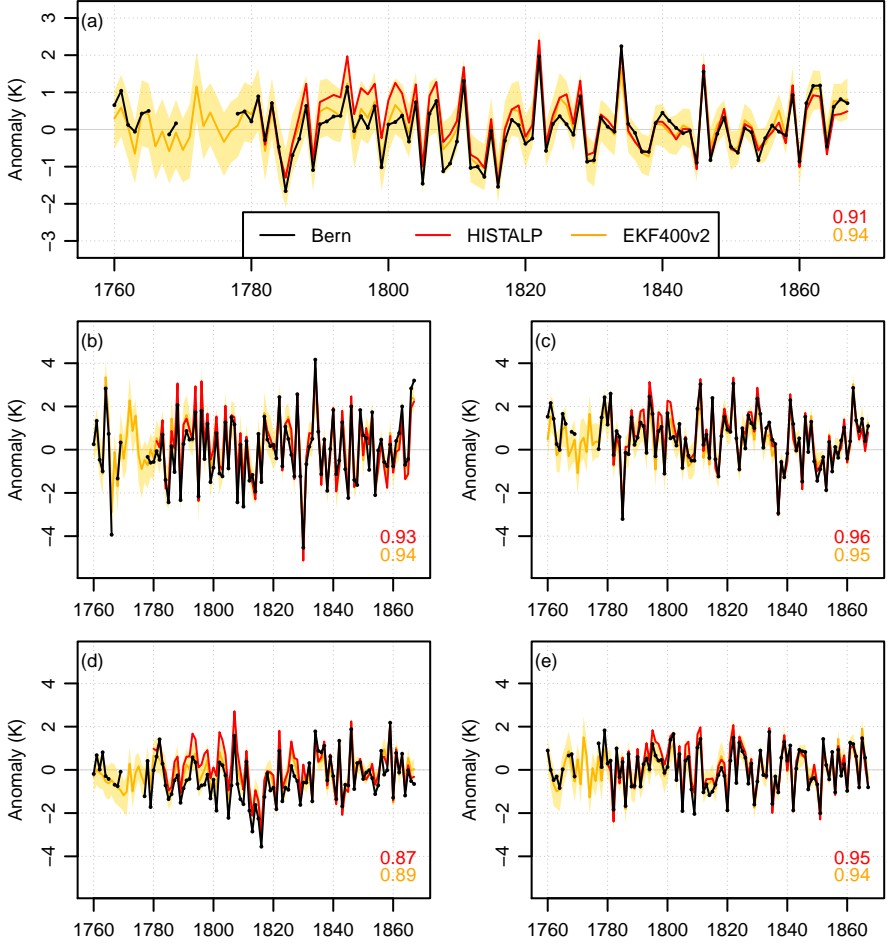

**Figure 11.** (a) Annual (December–November) temperature series for Bern and for the nearest grid point of the HISTALP and EKF400 data sets, expressed as anomalies with respect to the period 1801–1850. The shading represents the ensemble spread in EKF400. The same is also shown for (b) December–February, (c) March–April, (d) June–August, and (e) September–November. Pearson correlation coefficients between Bern and the other data sets are also given.

A similarly warm sequence occurred in the 1830s. At the other end of the spectrum, the coldest winter of the analyzed period (1829/1830) brought 46 cold days between 18 November and 7 February.

**4.4 Climate Variability Since the mid-18th Century**

When looking at the evolution of the annual mean temperature during the last 265 years (Fig. 14), one can recognize a weak positive trend during the 19th century, then accelerating particularly in the second half of the 20th century. In contrast, the HISTALP data set shows a strong negative trend over the 19th century.

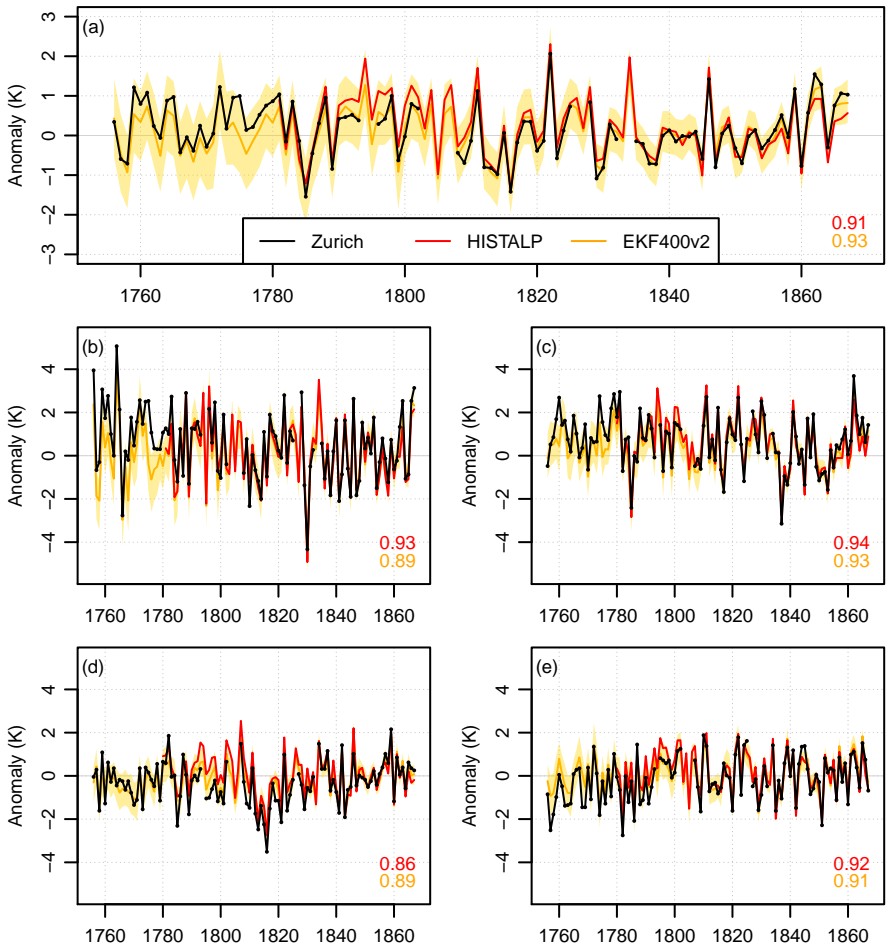

**Figure 12.** (a) Annual (December–November) temperature series for Zurich and for the nearest grid point of the HISTALP and EKF400 data sets, expressed as anomalies with respect to the period 1801–1850. The shading represents the ensemble spread in EKF400. The same is also shown for (b) December–February, (c) March–April, (d) June–August, and (e) September–November. Pearson correlation coefficients between Zurich and the other data sets are also given.

Our results for the mid-19th century, where our data diverge from HISTALP, are consistent with homogenized instrumental
series from northern Italy (Brunetti et al., 2006), with proxy-based reconstructions (e.g., Trachsel et al., 2012; Wetter and Pfister, 2013), and with Swiss glaciers variability (e.g., Zumbühl et al., 2008; Brönnimann et al., 2019b), all showing a strong positive trend (respectively a glacier retreat) between 1840–1870. This trend is also present in the homogenized series from Basel (Bider et al., 1958, Fig. 14), located in northern Switzerland, and in the EKF400 reconstruction (even though it assimilates the HISTALP data set).

Therefore, HISTALP is likely to have a significant warm bias before 1860 despite the additional corrections applied by Böhm et al. (2010). This bias is relevant for climate research as it affects many other widely used products that are based on





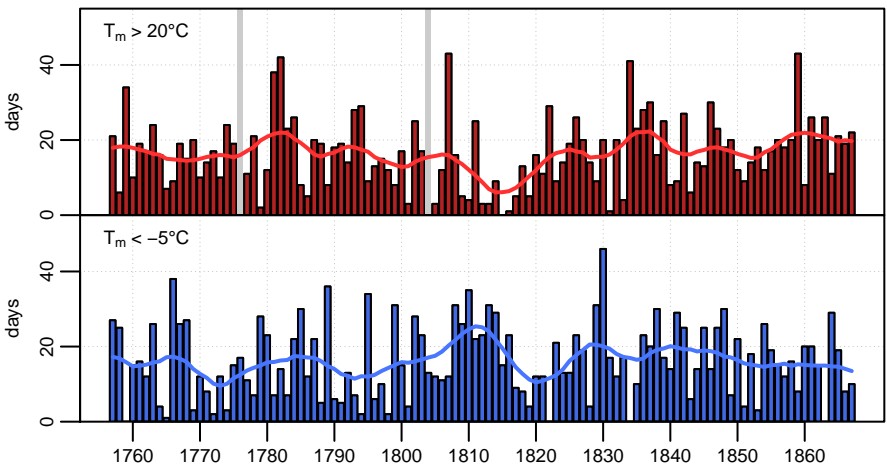

**Figure 13.** Number of days with mean temperature above 20°C in May–September (red) and below -5°C in November–March (blue) in the Swiss Plateau series. The indices are not calculated in years when more than 10% of days are missing (indicated by grey bars). The smoothed lines are produced using a Gaussian filter with $\sigma = 3$ years.

HISTALP data (e.g Rennie et al., 2014; Osborn et al., 2021). Moreover, HISTALP data are often used to calibrate and validate proxy-based reconstructions (Frank et al., 2007).

During most of the early instrumental period (i.e., before 1864) the Basel and EKF400 series are slightly warmer than the
Bern and Zurich series but remain well below HISTALP. There is remarkable agreement between the annual series of Bern and Zurich, with the exception of the 1830s–1840s, when Bern is up to 0.6 K warmer. The mean absolute difference between the two annual series is only 0.18 K before 1864. The 19th century (1801–1900) warming as estimated from a least-squares linear regression is 0.6 K in Bern and 0.7 K in Zurich; these trends are larger than in the Basel series (0.2 K).

The agreement between the Bern and Zurich series is not as good when looking at seasonal averages (April–September:
Fig. 15, October–March: Fig. 16), where Bern has higher anomalies than Zurich in the warm season (+0.61 K on average) and lower anomalies in the cold season (-0.49 K) during the early instrumental period. This is unlikely to be a true climate signal but is rather a measure of the uncertainty in the homogenization, in particular in the adjustments at the end of the early instrumental period.

The Swiss Plateau series matches relatively well the EKF400 reconstruction during the warm season, indicating that the
homogenization errors in the Bern and Zurich series roughly compensate each other. In winter EKF400 is warmer, which is expected because there are fewer assimilated proxies and, therefore, the influence of HISTALP is larger.

In Fig. 15 we also show the Alpine temperature reconstruction by Trachsel et al. (2012), based on tree-rings and lake proxies (note that the reconstruction is for the months of June to August only), while in Fig. 16 we show the cold season recostruction by Reichen et al. (2022), based on plant and ice phenology. Both agree best with the Swiss Plateau instrumental series, although
the winter reconstruction has a smaller trend over the 19th century.




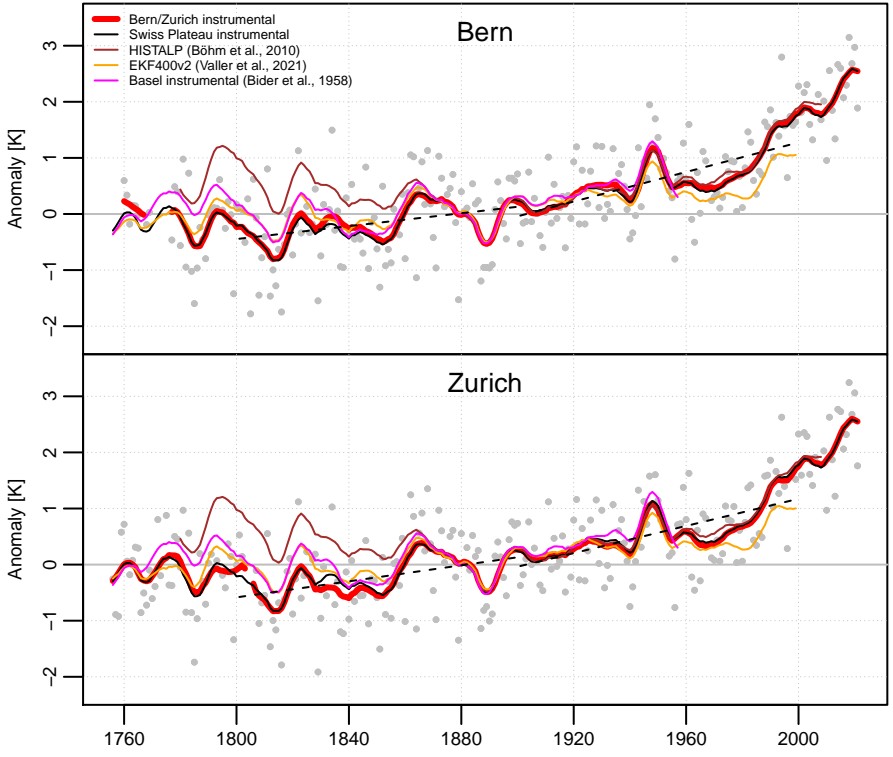

**Figure 14.** Smoothed time series of the annual (January–December) mean temperature anomalies (with respect to the 1871–1900 average) for Bern and Zurich (red thick lines), and their average (black line). Dots represent the unsmoothed annual anomalies. The brown and orange lines show the annual mean temperature from the nearest grid point in HISTALP and EKF400, respectively. The magenta line shows the annual mean temperature for Basel. The dashed lines show the linear trends for the 19th and 20th century in the Bern and Zurich series. All data series were smoothed using a Gaussian filter with $\sigma = 3$ years.

## 5   Conclusions

We provide two new long instrumental temperature series at daily resolution for Bern and Zurich, covering the period preceding the start of official measurements in Switzerland (1756–1863). Given the large heterogeneity and uncertainty of the underlying data, we also provide error estimates for each daily and monthly average. In addition, we merged the two series into a more complete series representing the central Swiss Plateau.

Some versions of early instrumental monthly temperature series for Bern and Zurich are already available in global data sets. We extended them further back in time (by over 70 years in the case of Zurich) and used more data sources. Being based on the raw sub-daily measurements, our results are independent from previous work, making them particularly valuable for assessing the quality of widely used long European records, many of which were reduced to daily or monthly means over a century ago.





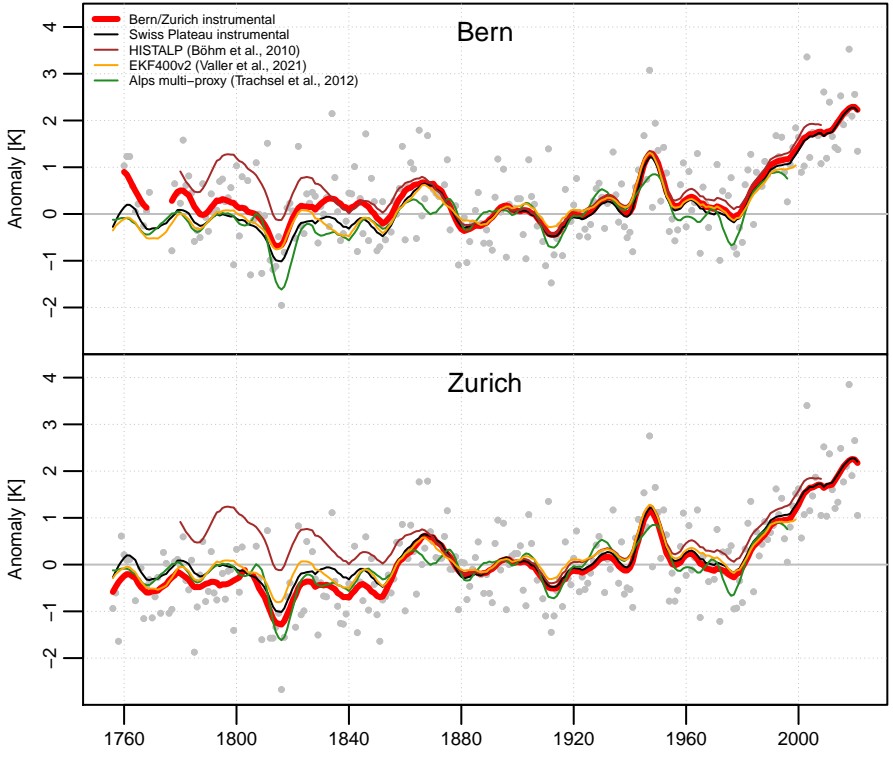

**Figure 15.** Smoothed time series of the warm season (April–September) mean temperature anomalies (with respect to the 1871–1900 average) for Bern and Zurich (red thick lines), and their average (black line). Dots represent the unsmoothed seasonal anomalies. The brown and orange lines show the annual mean temperature from the nearest grid point in HISTALP and EKF400, respectively. The green line represents the summer temperature "mean" reconstruction from multiple proxies for the Greater Alpine Region by Trachsel et al. (2012). All data series were smoothed using a Gaussian filter with $\sigma = 3$ years.

Moreover, each and every measurement that we use is fully traceable down to the original historical sources, which are, for the large majority, freely accessible from an online repository (Pfister, 2019).

    The data are homogenized taking advantage of the large number of early instrumental Swiss series that were digitized recently and making use of detailed metadata where available. Nevertheless, the large fragmentation of the data across dozens of observers and the frequent gaps made the homogenization particularly challenging. Therefore, we expect residual biases

to affect the results, particularly on seasonal and monthly scale. Additional reference series, for instance from neighboring countries, could improve data homogeneity especially in early years and during the transition to a NWS in the 1860s.

    The comparison with existing monthly temperature reconstructions allowed us to pinpoint problematic periods in the HISTALP data set, which is integrated in all global public instrumental databases and is widely used for calibration and validation of climate reconstructions for Central Europe. In particular, we pointed out a probable inhomogeneity around 1860,




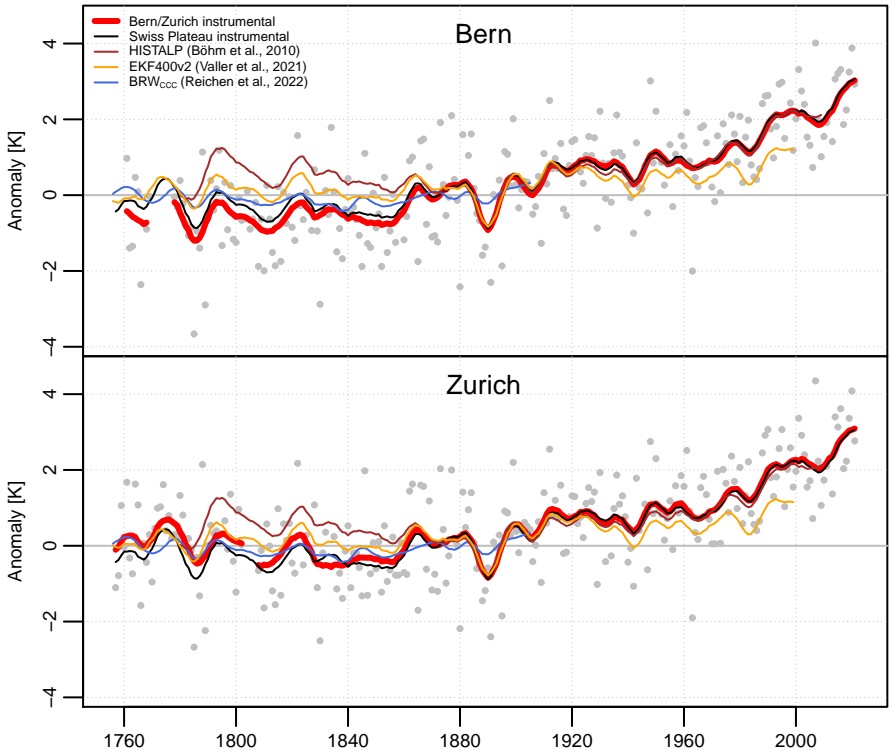

**Figure 16.** Smoothed time series of the cold season (October–March) mean temperature anomalies (with respect to the 1871–1900 average) for Bern and Zurich (red thick lines), and their average (black line). Dots represent the unsmoothed seasonal anomalies. The brown and orange lines show the annual mean temperature from the nearest grid point in HISTALP and EKF400, respectively. The blue line represents the nearest grid point of the cold season (October–May) reconstruction by Reichen et al. (2022). All data series were smoothed using a Gaussian filter with $\sigma = 3$ years.

which causes a positive bias for the whole early instrumental period, and an additional temperature overestimation between 1790–1805. The latter problem affects mainly the warm season.

   Our results suggest that pre-industrial climate in Switzerland was colder than previously thought. This highlights the still substantial uncertainty affecting climate variability in the early instrumental period – even for annual mean temperature in data-rich Central Europe – and points to the need for a revisitation of past homogenization efforts and for easier access to and 575  better traceability of the raw data.

*Data availability.* The daily and monthly raw and homogenized series will be made available through the BORIS data repository of the University of Bern (https://boris.unibe.ch/). The underlying raw sub-daily data will be available from MeteoSwiss and from the Copernicus Climate Change Service (C3S) Climate Data Store (https://cds.climate.copernicus.eu)



*Author contributions.* YB prepared and homogenized the instrumental data, performed the analysis, and wrote the manuscript. LP, CH,
and YB did the archive work and collected metadata. VV prepared the EKF400 data. SB supervised the work. All authors reviewed the
manuscript.

*Competing interests.* The authors declare that they have no competing interests.

*Acknowledgements.* This work was supported by GCOS Switzerland (project "Long Swiss Meteorological Series"), the Swiss National
Science Foundation (projects "CHIMES", grant no. 169676, and "REUSE", grant no. 162668), and the European Union (H2020; ERC grant
no. 787574 PALAEO-RA). The modern temperature data for Bern and Zurich have been provided by MeteoSwiss, the Swiss Federal Office
of Meteorology and Climatology. We thank the University Library Basel for providing digital images of several weather journals during the
COVID-19 pandemic, and the many students of the University of Bern who worked on the data keying.



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
