# Peer review of "Pre-industrial Temperature Variability on the Swiss Plateau Derived from the Instrumental Daily Series of Bern and Zurich"

_Climate of the Past, 2022_

## Author Response (AR1)

**Referee 1**

**Add kind of thermometer and observation times (when available) in table 1 and 2 for the different sub-periods, this will facilitate the reader.**

We added the kind of thermometer and the number of observations per day.

**The EIP warm bias of the HISTALP dataset is still an open problem: as the authors mention in their manuscript, also in Böhm et al. 2010 an important warm bias (up to 1°C in some seasons) with respect to other reconstructions (based on the same data, but undergoing different homogenization) is evident in the EIP (see figure 14 of the mentioned paper). The same warm bias is confirmed also by some proxy reconstructions (see e.g. Frank et al. 2007 Warmer early instrumental measurements versus colder reconstructed temperatures: shooting at a moving target. Quat Sci Rev 26:3298–3310). The ever-increasing availability of data for EIP is key to solving this dilemma.**

We added a sentence in the conclusions to emphasize this.

**This is not mandatory, but when the number of sub-daily observations allows it, I suggest the authors to extrapolate the minimum and maximum daily temperatures: their daily values will probably be affected by a high uncertainty, but their monthly averages could provide a relevant information. Also the availability of the daily temperature range at monthly resolution provides the user with a good instrument to improve the homogenization, the DTR being very sensitive to inhomogeneities.**

We appreciate the suggestion but we prefer to tackle Tmax and Tmin in a separate work. We have measurements from max/min thermometers from the 1820s onward that can be used for this, but they require significant additional work. Also the approach to the homogenization should probably be different (e.g., quantile matching). We would also like to produce daily pressure series in the future.

**Referee 2**

**Line8, it would be great to list the effected instrumental datasets, or at least a couple of the most used ones. Are HISTALP and EKF400 among those datasets? Again, it would be nice to provide a couple of field example as a demonstration of the significant implications.**

Unfortunately this is very hard to do, because it is very hard to figure out what is actually used in the various data sets and what the impact of each data source is. Therefore we can only speculate, knowing that HISTALP is very popular (for good reasons!). To better reflect this, we changed the words "biases affecting" to "biases that might affect".

**I think the describe in section 2 could be more compact, leaving out some of the personal information about the observer while focusing on information closely relevant to data availability and reliability. For example, what is the purpose of mentioning an observer lived in at least three different apartments (Line115)? Does it have an influence on the temperature measurements? If that is the case, it should be pointed out in the description.**

We tried to be as concise as possible, but given the large number of segments there is a lot of relevant information. We think that also the historical context and the motivation of the observers

are important. Where an observer lived is relevant as measurements were usually made at the observer's apartment. Nonetheless, we removed some non-essential information that can be found in the provided references.

**Line133-134, the two segments of temperature observations are not continuous, especially the first segment only covers three-year from December 1803 to November 1806 (with many gaps). So the subtitle "Fueter (1803-1833)" is kind unjustified and misunderstanding, to my view.**

The subsection's title is now "Fueter (1803–1806 and 1819–1833)".

**Line 354-360, please provide a brief rationale statement for the use of 90 vs. 45 vs. 30 minutes as the time error for the different types of observation.**

We added a few sentences in Sect. 3.2.3.

**Line453-454ï¼î"We introduced a subjective quality ranking (from 1, best, to 4, worst) based on our knowledge of the data and metadata (homogeneity, exposure, representativity, completeness, etc.)." Please provide more detail information about how the ranking has been done, maybe with an example, such as why and how were the three studer's observations (Table 1) given two different rankings. Also please describe how has the ranking (1-4) being applied in the reconstruction of temperature?**

We added a few sentences in Sect. 3.4.

**The subtitle of section4.4 is about "Climate Variability Since the mid-18th Century", however, I do not see any mention of the results during the early period of 1760-1800. A paragraph discussion of this period would be valuable and also fulfill the subtitle.**

We added a paragraph in Sect. 4.4.

**Line 557-559 "Being based on the raw sub-daily measurements, our results are independent from previous work, making them particularly valuable for assessing the quality of widely used long European records, many of which were reduced to daily or monthly means over a century ago." I am a little confused about what does the last phrase of the sentence mean. Is "which" mean the resolution of the "widely used long European records"? And that these records have daily or monthly resolutions before 20th century?**

We rephrased the sentence.

**Figure 1. Does the size of the red circles indicate the number of stations in each region? Please provide a notation.**

No, the circles on Bern and Zurich have a larger size simply to highlight their position. We mentioned it in the caption. In addition, we improved the map's quality by adding the topography.

**Some of the lines in Fig. 14-16 are difficult to distinguish from one another. Please consider revise the figures using more easily distinguished colors.**

We now use colorblind-friendly color scales in these figures. Also, another level of line thickness was added to improve readability.

In addition to the above points, we made the following changes to the manuscript:

- **Section 4**: There are minor changes in the results after including better reference series from Basel and Geneva that were produced within the same project (a companion paper is in preparation). These changes do not affect our conclusions.

- **Code and data availability**: Information on how to access the raw and processed data as well as the homogenization code has been added.

- Verb tenses were made more consistent throughout the manuscript.

Kind regards,

Yuri Brugnara